# Pretreatment Strategies to Enhance Enzymatic Hydrolysis and Cellulosic Ethanol Production for Biorefinery of Corn Stover

**DOI:** 10.3390/ijms232113163

**Published:** 2022-10-29

**Authors:** Wan Sun, Xuezhi Li, Jian Zhao, Yuqi Qin

**Affiliations:** 1National Glycoengineering Research Center, Shandong University, Qingdao 266237, China; 2State Key Laboratory of Microbial Technology, Shandong University, Qingdao 266237, China

**Keywords:** pretreatment, enzymatic hydrolysis, fed-batch semi-simultaneous saccharification and fermentation, ethanol, corn stover

## Abstract

There is a rising interest in bioethanol production from lignocellulose such as corn stover to decrease the need for fossil fuels, but most research mainly focuses on how to improve ethanol yield and pays less attention to the biorefinery of corn stover. To realize the utilization of different components of corn stover in this study, different pretreatment strategies were used to fractionate corn stover while enhancing enzymatic digestibility and cellulosic ethanol production. It was found that the pretreatment process combining dilute acid (DA) and alkaline sodium sulfite (ASS) could effectively fractionate the three main components of corn stover, i.e., cellulose, hemicellulose, and lignin, that xylose recovery reached 93.0%, and that removal rate of lignin was 85.0%. After the joint pretreatment of DA and ASS, the conversion of cellulose at 72 h of enzymatic hydrolysis reached 85.4%, and ethanol concentration reached 48.5 g/L through fed-batch semi-simultaneous saccharification and fermentation (S-SSF) process when the final concentration of substrate was 18% (*w*/*v*). Pretreatment with ammonium sulfite resulted in 83.8% of lignin removal, and the conversion of cellulose and ethanol concentration reached 86.6% and 50 g/L after enzymatic hydrolysis of 72 h and fed-batch S-SSF, respectively. The results provided a reference for effectively separating hemicellulose and lignin from corn stover and producing cellulosic ethanol for the biorefinery of corn stover.

## 1. Introduction

The increasing demand for energy and deficiency in fossil fuels aggravates energy security and environmental problems throughout the world [1,2]. Bioethanol has been regarded as an alternative to fossil fuel. Bioethanol can be added directly to gasoline, which is widely used in many countries; for example, Brazil currently uses the E27 on all transport vehicles and the E100 on flex-fuel vehicles [3]. It is expected that global bioethanol production will be increased by 20% to 130 billion liters in 2024 [3]. Now, the development of the first-generation bioethanol from grain has been limited. The second-generation bioethanol, which was produced using lignocellulosic biomass as raw materials, has been extensively studied in recent decades because of renewable, abundant, and low-cost raw materials, eco-friendly conversion process, etc. [4,5,6]. The lignocellulose biomass mainly includes agricultural wastes such as wheat straw, corn stover, sugarcane bagasse, etc., and forestry processing residues. Corn stover, as an agricultural residue, is one of the most abundant renewable lignocellulose and has been widely studied for bioethanol production [4,7,8]. Corn stover usually contains 30–35% of cellulose, 19–22% of hemicellulose, and 18–22% of lignin [9]. The three components are associated with each other by complex bonds to form a recalcitrant structure that limits the enzymatic hydrolysis of cellulose. Among them, lignin is an aromatic polymer macromolecule consisting of three different phenolic monomers which cannot be used for bioethanol fermentation [10,11]. It has been reported that lignin is the main obstacle to enzymatic hydrolysis of cellulose because it could hinder cellulase access to the cellulosic substrate as a steric hindrance while nonproductively absorbing cellulase, resulting in the loss of enzyme activity [12,13]. Pretreatment must be first carried out to overcome the biomass recalcitrance before enzymatic hydrolysis and fermentation [14] to improve the enzymatic digestibility of lignocellulose and ethanol yield. Generally, there are three main steps for the bioconversion of lignocellulose into bioethanol: pretreatment, enzymatic hydrolysis, and ethanol fermentation [5]. 

Various pretreatment methods have been tested for improving the enzymatic hydrolysis of corn stover [5,10,15]. It was reported that dilute acid pretreatment could effectively remove hemicellulose, thereby improving the accessibility of enzymes to cellulose and enhancing the enzymatic digestibility of corn stover [16,17,18,19,20,21,22]. Various organic and inorganic acids have been used for the acidic pretreatment of corn stover, for example, sulfuric acid, phosphoric acid, acetic acid, peracetic acid, and maleic acid, etc. [20,21,22,23,24]. Among these, sulfuric acid is the most commonly used acid because of its good ability to effectively remove hemicellulose and its low cost. For example, Sievers et al. [21] used dilute sulfuric acid to treat corn stover and obtained the highest xylose yield of 80% at 165 °C for 10 min with 1% H_2_SO_4_. Dilute acid pretreatments with low severity could achieve high conversion of xylan to xylose, which is beneficial to achieve favorable overall process economics. However, the dilute acid pretreatment had little influence on lignin, thus high content of lignin was retained in pretreated corn stover. Most literature reported that lignin had a great adverse impact on enzymatic hydrolysis through a steric hindrance to physically limit cellulase access to cellulose and through nonproductive adsorbing of enzymes, which leads to enzyme activity loss [25]. Research has reported that lignin removal was an effective method for improving enzymatic hydrolysis performance. Some methods have been investigated for removing lignin in lignocellulose; for example, NaOH pretreatment [26,27], organic solvent pretreatment with formic acid or acetic acid [28], and sulfite pretreatment [29,30,31,32]. The sulfite pretreatment was regarded as a potential process for improving the enzymatic hydrolysis efficiency of lignocellulose and has been used to pretreat corncob [29], softwood [33], oil palm empty fruit bunch [32], and so on. During sulfite pretreatment, lignin is largely sulfonated by the sulfite ion groups (such as SO_3_^2−^, and HSO_3_^−^), which results in higher hydrophilicity. In addition, sulfate pretreatment can be performed over a wide range of pH and temperature to selectively remove lignin or hemicellulose by adjusting the pH of the pretreatment liquid [33,34]. Zhang D.S. et al. [35] reported that sulfite pretreatment to overcome the recalcitrance of lignocelluloses (SPORL) partially dissolved lignin in Switchgrass because of the improvement of lignin hydrophilicity through sulfonation. Zhu et al. [36] also reported that SPORL solubilized approximately 45% of lignin in wood. In the sulfite pretreatment, the addition of alkali could enhance lignin removal through the combined action of alkali dissolution and sulfonation [37,38]. For example, Li et al. [38] reported that lignin removal of about 92% could be achieved by an alkaline sodium sulfite (ASS) pretreatment at a relatively low temperature of 140 °C with 5% NaOH and 5% Na_2_SO_3_ (based on the dry weight of biomass) for corn stover. Additionally, in pulping industry, ammonium sulfite (AS) was used to remove lignin for a long time, but only a few studies were reported on the pretreatment of lignocellulosic biomass with ammonium sulfite [31]. Qi et al. [39] used AS to treat wheat straw and found that pretreatment under severe conditions could facilitate delignification and enzymatic hydrolysis. It is worth mentioning that the black liquid of AS pretreatment could be used as fertilizer for agriculture [40], which decreases wastewater treatment costs and environmental pollution. Thus, as a good potential pretreatment process, the feasibility of applying AS pretreatment to improve enzymatic hydrolysis of corn stover still needs to be further investigated.

On the other hand, most of the previous studies about dilute acid pretreatment and sulfite pretreatment of lignocellulose mainly focused on how to improve the enzymatic conversion of cellulose in corn stover and cellulosic ethanol production, particularly on the conversion of a single component in corn stover such as cellulose. However, little attention was paid to the whole application of different chemical components in corn stover. It has been testified that hemicellulose in lignocellulose is easy to be degraded by an acidic pretreatment process, and lignin could be removed by an ASS process. Thus, it was expected to effectively fractionate cellulose, hemicellulose, and lignin of corn stover through a two-step process combining DA pretreatment and ASS pretreatment, which was helpful for realizing utilization of different components and biorefinery. Based on this, in this study, we tried to use two pretreatment strategies. One was a joint process combining DA and ASS to achieve fractionation of three main components of corn stover for recovering xylose (a bulk chemical) and sulfonated lignin that could be used as a surfactant and concrete water reducer, etc. The other was AS pretreatment for removing lignin, which could be used as fertilizer in agriculture. The conditions of pretreatments, such as reaction temperature, chemical dosage, and solid-to-liquid ratio, were optimized for maximizing the hemicellulose sugar yield and lignin removal rate, respectively. The pretreated corn stover with high cellulose content was further enzymatically hydrolyzed and fermented into cellulosic ethanol by a fed-batch S-SSF process for assessing the feasibility of ethanol production using the two pretreatment strategies. This study will provide a reference for cellulosic ethanol production from corn stover and fractionating chemical compositions for biorefinery of corn stover.

## 2. Results

### 2.1. Pretreatment Process Optimization Combining DA and ASS for High-Efficiency Separating Xylose and Lignin and Improving Enzymatic Digestibility of Corn Stover

#### 2.1.1. Effect of DA Pretreatment

The changes in contents of glucan, xylan, and lignin in corn stover before and after DA pretreatment under different conditions are listed in Table 1. Xylan in biomass can be relatively easily removed by DA pretreatment. As shown in Table 1, with the increase in temperature, acid dosage, and the decrease in solid-to-liquid ratio, the content of xylan in pretreated corn stover gradually decreased. When the temperature continued to rise to 150 °C, and the solid-to-liquid ratio was 1:10, the xylan content of corn stover was reduced to about 3% after DA pretreatment under different acid concentrations. However, the xylose recovery in the pretreated liquid showed different and more complex changes. Reducing the solid/liquid ratio improved the xylose yield. At low acid dosage (0.8–1.0%), the xylose recovery increased with the increase in pretreatment temperature, while high acid dosage (1.2%) and higher temperatures resulted in lower xylose recovery. At the same conditions of temperature and solid-liquid ratio, higher acid dosage decreased the recovery of xylose, especially at 150 °C. The highest xylose recovery of 93% was obtained under the conditions of 150 °C, 0.8% of acid dosage, and a 1:10 (*w*/*v*) ratio of solid to liquid. Table 1 also gives the contents of hydroxymethylfurfural (HMF) and furfural, the degradation products of glucose and xylose, respectively, in the dilute acid pretreatment solution. It was also shown that the contents of furfural and HMF increased with increasing dilute acid pretreatment severity (increasing pretreatment temperature or increasing acid dosage). On the whole, dilute acid pretreatment effectively removed the xylan in the corn stover, but more lignin remained in the pretreated corn stover. The high lignin content in pretreated corn stover affected the enzymatic digestibility of pretreated corn stover, and the conversion of glucan in pretreated corn stover reached about 70% (Table 1). In order to further improve enzymatic digestibility, it is necessary to further remove lignin.

#### 2.1.2. Effect of ASS Pretreatment

It was reported by many studies that alkali-based pretreatment could remove lignin in lignocellulose [26,27,41,42]. Li et al. [38] reported that the addition of Na_2_SO_3_ in alkali-based pretreatment improved the recovery of carbohydrates and the removal of lignin. In this study, ASS pretreatment was further used to remove lignin in DA-pretreated corn stover, and the effects of pretreatment conditions, including NaOH dosage (on the oven-dry weight of raw material), Na_2_SO_3_ dosage (on the oven-dry weight of raw material), solid-to-liquid ratio, reaction temperature on lignin removal, glucan recovery, and enzymatic digestibility were investigated. It can be clearly seen in Figure 1 that ASS pretreatment can significantly remove lignin and improve the enzymatic digestibility of pretreated corn stover. To study the effects of Na_2_SO_3_ dosage, Na_2_SO_3_ of 9%, 12%, 15%, and 18% (on the oven-dry weight of DA-pretreated corn stover) was used in the ASS pretreatment, respectively, and the pretreatments were carried out at 140 °C with 5% NaOH and the solid-to-liquid ratio of 1:6 for 1 h.

As shown in Figure 1a, the lignin removal significantly increased with sodium sulfite dosage increase from 9% to 18%. Initially, the glucan recovery increased slowly (when Na_2_SO_3_ dosage < 12%), then decreased with the increasing dosage of Na_2_SO_3_. The maximum removal of lignin reached 85% at 15% Na_2_SO_3_, while the glucan recovery was 83.7%. Figure 1b shows that under the conditions of 12% Na_2_SO_3_, the solid-to-liquid ratio of 1:6, and 160 °C for 1 h, increasing the NaOH dosage promoted lignin removal and improved the enzymatic digestibility of corn stover, but resulted in lower glucan recovery. According to Figure 1c, although the highest lignin removal rate (85.9%) was obtained at the solid-to-liquid ratio of 1:8, 12% Na_2_SO_3_, 5% NaOH, and 150 °C for 1 h, there was a higher glucan conversion rate (80.5%) at 72 h of enzymatic hydrolysis of pretreated corn stover and glucan recovery (92.7%) at the 1:6 of solid-to-liquid ratio compared with that at the 1:8 of solid-to-liquid ratio (76.4% for glucan conversion and 88.3% for glucan recovery, respectively). Figure 1d shows that there were no significant differences in glucan recovery, lignin removal, and glucan conversion under different pretreatment temperatures.

### 2.2. Effect of Ammonium Sulfite (AS) Pretreatment on Lignin Removal and Enzymatic Digestibility of Corn Stover

As shown in Table 2, AS pretreatment can effectively remove lignin in corn stover and improve the enzymatic digestibility of lignocellulose, but the AS dosage and the solid-to-liquid ratio during pretreatment affected the pretreatment performance. At the same solid-to-liquid ratio of 1:6, increasing the dosage of ammonium sulfite enhanced the lignin removal, and the contents of glucan and xylan in pretreated corn stover increased. When the dosage of ammonium sulfite was 14%, the solid-to-liquid ratio decreased from 1:2 to 1:6, the lignin content decreased from 8.4% to 3.7%, and the lignin removal rate increased from 55.8% to 83.8%, but with further change in the solid-to-liquid ratio to 1:8, the lignin removal changed slightly.

For the glucan conversion, it was found in Table 2 that when the dosage of ammonium sulfite was 10%, the conversion of glucan reached 88.0%, and the xylan conversion rate reached 99.8%. Further increasing ammonium sulfite dosage did not result in a higher glucan conversion. The solid-to-liquid ratio in AS pretreatment had a significant effect on the enzymatic digestibility of AS-pretreated corn stover. When the ratio of solid-to-liquid was changed from 1:2 to 1:6, the conversion rate of glucan increased from 70.3% to 87.4%.

### 2.3. Fed-Batch S-SSF of Pretreated Corn Stover for Ethanol Production

#### 2.3.1. Fed-Batch S-SSF of DA-ASS-Pretreated Corn Stover

Figure 2 gives the changes in substrate concentration, glucose concentration, ethanol concentration, and ethanol yield during the fed-batch S-SSF of DA-ASS-pretreated corn stover. Here, 0 h in Figure 2 was the time that yeast was inoculated to the reaction system. The initial substrate concentration in fermentation was 13%, and feeding was carried out at different time points according to the liquefaction of the reaction system, and the final concentration of substrate reached 18.1%. It was found that the glucose produced by enzymatic hydrolysis can be quickly converted into ethanol, and after 138 h of fermentation time, the glucose concentration during the fermentation was always lower than 0.1 g/L. At 163 h of fermentation, the ethanol concentration reached 48.5 g/L, corresponding to a 77.2% of ethanol yield.

#### 2.3.2. Fed-Batch S-SSF of AS-Pretreated Corn Stover

As there were large amounts of glucan and xylan in AS-pretreated corn stover, glucose and xylose were simultaneously released during enzymatic hydrolysis of AS-pretreated corn stover. We used two strains, *S. cerevisiae* XH7, capable of metabolizing glucose and xylose [43], and Angel *Saccharomyces cerevisiae,* capable of just metabolizing glucose as a comparison, to assess the feasibility of ethanol production through fed-batch S-SSF process using the AS-pretreated corn stover. Figure 3a shows that Angel *S. cerevisiae* can rapidly consume glucose to produce ethanol. The residual glucose concentration in liquid after 60 h of fermentation was lower than 2 g/L, and the ethanol concentration reached 45 g/L when the final concentration of substrate was 18%, and the ethanol yield was 71.6%. However, it was also shown that xylose cannot be utilized by Angel *Saccharomyces cerevisiae* and was mostly retained in the fermentation liquid. Figure 3b shows that *S. cerevisiae* XH7 could consume both glucose and xylose to produce ethanol; glucose was preferentially consumed, and xylose utilization was slower than glucose utilization. At 60 h of fermentation, the concentrations of glucose and xylose were maintained at a low level, and the ethanol concentration reached 50 g/L, corresponding to 64.7% of ethanol yield (based on total sugar).

## 3. Discussion

DA pretreatment can realize the recovery of hemicellulose such as xylose, but xylose and glucose can be further degraded, resulting in the production of furfural and HMF that inhibit subsequent ethanol fermentation [44,45]. Thus, suitable DA pretreatment severity was necessary for increasing sugar recovery and decreasing inhibitor formation. On the other hand, most of the lignin remained in the pretreated lignocellulose after DA pretreatment, and the presence of lignin can physically hinder cellulase accessible to cellulose or cause nonproductive adsorption of cellulase onto lignin, thereby affecting the enzymatic hydrolysis of DA-pretreated corn stover [46]. Our results showed that, after DA pretreatment, the conversion of glucan to glucose was only about 70%. To achieve a high enzymatic hydrolysis performance while separating lignin for further application, alkali-based pretreatment was further used to remove lignin in DA-pretreated corn stover. It was reported that sodium hydroxide could effectively dissolve lignin, and sodium sulfite can improve the solubility of lignin through a sulfonation reaction between SO_3_^2−^ and lignin [38,47]. This study showed that a high NaOH dosage enhanced lignin removal but lowered glucan recovery because of the cellulose dissolution under alkali conditions during ASS pretreatment. At the set chemical dosages, the change in the solid-to-liquid ratio influenced the chemical (NaOH and Na_2_SO_3_) concentrations in the pretreatment liquid, which also affected pretreatment performance. However, there was no significant difference in the glucan conversion rate after ASS pretreatment under different experiments conditions, which may be due to the fact that lignin content was no longer the main factor influencing enzymatic hydrolysis of lignocellulose when lignin in lignocellulose was removed to a certain extent (for example, above 70% of lignin was removed after ASS pretreatment in this study). After the combining pretreatment of DA and ASS, the conversion of glucan in pretreated corn stover can reach more than 80%, indicating that the enzymatic digestibility of corn stover was effectively improved. Moreover, using the combining pretreatment of DA and ASS, the xylan and lignin can be respectively effectively separated from corn stover for their further application; for example, in this study, 93% of xylan was recovered in the low-severity DA pretreatment step (with 0.8% H_2_SO_4_ at 150 °C), and 85.9% of lignin was removed in the second ASS step using 12% Na_2_SO_3_ and 5% NaOH (based on the dry weight of corn stover) at the solid-to-liquid ratio of 1:8, 150 °C for 1 h, while the conversion of cellulose reached 80% above by enzymatic hydrolysis. Lee et al. [22] used a sequential dilute acid and alkali pretreatment process to treat corn stover and found that about 74.6–77.3% of xylan in corn stover was hydrolyzed by 0.5–1% H_2_SO_4_ at 140–180 °C, and 85.9–89.4% of lignin was removed under the conditions of 20% NaOH, 80 °C, and the solid-to-liquid ratio of 1:20 for 1 h in the second pretreatment step. However, a shortcoming of the DA and ASS combining process was the long technological process, resulting in high devices cost and long operation times.

It was reported that ammonium sulfite could also remove lignin from lignocellulose through ammonolysis and sulfonation [30]. In this process, waste pretreatment liquid containing residual ammonium sulfite after pretreatment can be used as a fertilizer for agricultural production, and sulfonated lignin could be used as a surfactant and concrete water-reducer, which could increase the economic benefits of the whole process for biorefinery of corn stover and reduce the impact on the environment. During the ammonium sulfite pretreatment, the addition of ammonium carbonate can stabilize the pH, thus reducing the degradation of hemicellulose. By optimizing conditions of ammonium sulfite pretreatment, in this study, it was proven that the lignin could be effectively removed from raw corn stover, and high conversions of glucan and xylan were obtained in subsequent enzymatic hydrolysis.

To achieve economic feasibility, it is essential for saccharification and fermentation at high substrate concentrations to obtain a high concentration of final products such as ethanol. Many studies have shown that exorbitant initial substrate concentration would lead to difficulty in heat transfer and mass transfer [48,49], which was not in favor of enzymatic saccharification and fermentation. The fed-batch fermentation process, however, could avoid the above problems [50,51,52]. Compared with the simultaneous saccharification and fermentation (SSF) process, Semi-SSF has the advantages of separate hydrolysis and fermentation (SHF) process and SSF process. In semi-SSF, a pre-hydrolysis step was firstly conducted prior to SSF, and in this stage, the cellulosic substrate could be highly efficiently degraded because of the high enzymatic activities of cellulase. At a suitable temperature of 50 °C, the solid substrate was liquefied, and the viscosity in the reaction system was rapidly reduced, while some glucose was produced in the prehydrolysis step, which was also conducive to the initial growth of yeast at the beginning of SSF. After the prehydrolysis, the temperature of the reaction system decreased to 30 °C for conducting subsequent SSF [53]. Using the fed-batch S-SSF process in this study, the feasibility of ethanol production using the corn stover pretreated with two pretreatment strategies as substrates was further assessed. As there were large amounts of glucan and xylan in AS-pretreated corn stover, high concentrations of glucose and xylose were released. It was known that glucose could be readily utilized by yeast as a common fermentable sugar, while xylose cannot be metabolized by ordinary yeast for ethanol fermentation [43]. *S. cerevisiae* XH7 is a rational metabolically engineered strain that rapidly utilizes xylose for fermentation [34]. In this study, we used two types of yeasts, Angel *S. cerevisiae* for fermentation of glucose from the corn stover pretreated with DA and ASS combining pretreatment, and *S. cerevisiae* XH7 for co-fermentation of glucose and xylose released from AS-pretreated corn stover, and Angel *S. cerevisiae* was used as a control. It was found that residual glucose was very low after fermentation, indicating that glucose was effectively used by the two yeasts. The final ethanol concentration reached 48.5 g/L for the DA and ASS combined pretreated corn stover and 50 g/L for AS-pretreated corn stover when substrate content was fed into 18% during S-SSF, respectively, which meet the requirements of industrial production for ethanol concentration (generally, higher than 4% (*v*/*v*)).

Based on the above experiments, as an example, Figure 4 gives the sketch of the biorefinery of corn stover by using the DA-ASS joint pretreatment strategy and the preliminary mass balance of the overall process of bioconversion. It was calculated that about 226 kg of xylose and 98 kg of ethanol could be obtained from 1000 kg raw corn stover (non-pretreated corn stover) using the DA and ASS joint pretreatment and fed-batch S-SSF process. The lignin fractions dissolved in liquid during ASS pretreatment could be recovered using an existing industrial process. In our previous study, hemicellulose sugar and lignosulfonate in liquid from bisulfite pretreatment of EFB (empty fruit bunch) from oil palm have been separated effectively using a macroporous resin DM130 and 95% methanol through resin–eluent sequence [54]. The lignin could be sold as a commodity and, for example, used as a surfactant and concrete water reducer, etc. Additionally, the solid residues from ethanol fermentation were mainly composed of lignin and could be prepared into lignin products or further converted into other chemicals, which has been investigated preliminarily in our previous study [55].

## 4. Materials and Methods

### 4.1. Materials and Strains

The corn stover used in this work was harvested in Jinan, Shandong province, China. After being cut to 10 cm by an industrial-scale grinder, the corn stove was stored in a sealed bag for pretreatment.

Cellulase with a filter paper activity (FPA) of 160 IU/g, β-glucosidase activity of 75 IU/g, xylanase activity of 12351 IU/g, and xylosidase activity of 30 IU/g were obtained from Baiyin Sainuo Technology, Ltd. (Baiyin, Gansu Province, PR China)

*Saccharomyces cerevisiae* XH7 was provided by Professor Xiaoming Bao from Qilu University of Technology, Shandong. The strain XH7 can utilize both glucose and xylose for fermentation. Angel *Saccharomyces cerevisiae* was purchased from Angel Yeast Co., Ltd (Yichang, Hubei, China).

### 4.2. Pretreatment of Corn Stover

The pretreatment was conducted in a rotary electrothermal pressure digester with four stainless steel reactors of 1.5 L and an electrical heating system. The single-factor experiment was used to optimize suitable pretreatment conditions, including temperature, chemical dosage, and solid-to-liquid ratio in the DA, AS, and ASS processes, respectively. Corn stover, chemicals, and water were added according to a certain proportion and reacted for a required time at the set temperature. After pretreatment, the spent liquid was separated from the solid through a filter cloth. To determine xylose content in the spent liquid, the pretreatment liquid was firstly neutralized with barium hydroxide and then centrifuged at 10,000 rpm for 10 min. The supernatant was properly diluted and filtered through 0.2 µm filters and detected by HPLC equipped with a refractive index detector and an Aminex HPX-87H column (BioRad, Hercules, CA, USA). The mobile phase was 5 mM H_2_SO_4_ at 40 °C with a flow rate of 0.6 mL/min. The separated solid (pretreated corn stover) was washed with tap water and then stored in plastic bags at 4 °C until used.

### 4.3. Enzymatic Hydrolysis

To evaluate the enzymatic digestibility of pretreated corn stover, the Enzymatic hydrolysis was carried out in a 100 mL flask with 50 mL of reaction volume containing pretreated substrate of 1 g (on dry weight) at pH 4.8 (0.05 M sodium citrate buffer), 50 °C, and 150 rpm for 72 h. The cellulase dosage was 20 FPU/g dry matter. Hydrolysates were taken out periodically, boiled for 10 min, and centrifuged at 10000 rpm for 10 min. The supernatant was analyzed for glucose content using the SBA-40C biological sensor analyzer (Institute of Biology, Shandong Academy of Sciences, Jinan, China).

### 4.4. Fed-Batch Semi-Simultaneous Saccharification and Fermentation (Fed-Batch S-SSF)

Moderate pretreated corn stover was sterilized in a 100 mL conical flask, then cellulase (20 FPU/g DM) and sodium citrate buffer (0.05 M, pH 4.8) were added to achieve prehydrolysis at 50 °C, at 150 rpm for 8 h for pretreated substrate with DA and ASS, and 12 h for pretreated substrate with AS. The initial substrate concentration was adjusted with sodium citrate buffer, 0.45 g yeast was activated for 2 h using 30 mL of 0.25% sterilized glucose solution at 30 °C, 150 rpm, and 0.5% of the activated yeast was inoculated into the reaction system for conducting subsequent SSF at 35 °C and 200 rpm. During fermentation, the moderate sterilized pretreated substrate was added into the reaction system in batches until the substrate concentration reached the setting concentration. The reaction system was sampled periodically. The contents of glucose, xylose, and ethanol were analyzed by HPLC equipped with a refractive index detector and an Aminex HPX-87H column (BioRad, Hercules, CA, USA). The mobile phase was 5 mM H_2_SO_4_ at 40 °C with a flow rate of 0.6 mL/min.

All the experiments were performed in triplicate. Mean value and standard variance were calculated using Origin 2022 and are shown in the paper.

### 4.5. Analytical Methods and Calculations

Chemical compositions of corn stover and pretreated corn stover, for example, contents of cellulose, xylan, lignin, extractives, etc., were determined according to analytical methods provided by the National Renewable Energy Laboratory (NREL) [56,57]. Briefly, the dried and pulverized corn stalks with sizes of 20 to 80 meshes were used for chemical component determination. The material was first extracted with ethanol, subjected to two-step acid hydrolysis (with 72% sulfuric acid at 30 °C for 1 h and 4% sulfuric acid at 121 °C for 1 h), and then filtered with a crucible. Finally, the contents of cellulose, hemicellulose, and lignin were determined, respectively.

To determine the contents of furfural and hydroxymethylfurfural (HMF) in pretreatment liquid, an aliquot (3 mL) of the pretreatment liquid was adjusted by barium hydroxide to pH 2 and then centrifuged at 10,000 rpm for 10 min. The supernatant was properly diluted and filtered through 0.2 µm filters, then analyzed by HPLC equipped with a UV detector and an Aminex HPX-87H column (BioRad, Hercules, CA, USA) at a wavelength of 215 nm (Shimadzu). The mobile phase was 5 mM H_2_SO_4_ at 40 °C with a flow rate of 0.5 mL/min [58].

Xylose recovery and lignin removal in the pretreatment stage and glucan conversion in enzymatic hydrolysis were calculated according to formula (1) to formula (4), respectively [4].
(1)$$\mathrm{Xylose}\;\mathrm{recovery}\;(\%)=\frac{\mathrm{xylose}\;\mathrm{in}\;\mathrm{pretreated}\;\mathrm{liquid}\;(g)}{\mathrm{xylan}\;\mathrm{in}\;\mathrm{raw}\;\mathrm{corn}\;\mathrm{stover}\;(g)\times 1.14}\times 100$$
(2)$$\mathrm{Lignin}\;\mathrm{removal}\;(\%)=\frac{\mathrm{lignin}\;\mathrm{in}\;\mathrm{raw}\;\mathrm{corn}\;\mathrm{stover}\;\left(g\right)-\mathrm{lignin}\;\mathrm{in}\;\mathrm{pretreated}\;\mathrm{corn}\;\mathrm{stover}\;(g)}{\mathrm{lignin}\;\mathrm{in}\;\mathrm{raw}\;\mathrm{corn}\;\mathrm{stover}\;(g)}\times 100$$



(3)
$$\mathrm{Glucan}\;\mathrm{conversion}\;(\%)=\frac{\mathrm{glucose}\;\mathrm{released}\;\mathrm{in}\;\mathrm{enzymatic}\;\mathrm{hydrolysis}\;(g)\times 0.9}{\mathrm{glucan}\;\mathrm{in}\;\mathrm{pretreated}\;\mathrm{corn}\;\mathrm{stover}\;(g)}\times 100$$


(4)
$$\mathrm{Xylan}\;\mathrm{conversion}\;(\%)=\frac{\mathrm{xylose}\;\mathrm{released}\;\mathrm{in}\;\mathrm{enzymatic}\;\mathrm{hydrolysis}\;(g)\times 0.88}{\mathrm{glucan}\;\mathrm{in}\;\mathrm{pretreated}\;\mathrm{corn}\;\mathrm{stover}\;(g)}\times 100$$



For AS-pretreated corn stover, ethanol yield after S-SSF was calculated by the Formula (5) [59]:(5)$$\mathrm{Ethanol}\;\mathrm{yield}\;(\%)=\frac{\mathrm{ethanol}\;\mathrm{in}\;\mathrm{fermentation}\;\mathrm{liquid}\;(g)}{\left[\mathrm{xylan}\;\mathrm{in}\;\mathrm{pretreated}\;\mathrm{corn}\;\mathrm{stover}\;\left(g\right)\times 1.14+\mathrm{glucan}\;\mathrm{in}\;\mathrm{pretreated}\;\mathrm{corn}\;\mathrm{stover}\left(g\right)\times 1.11\right]\times 0.51}\times 100$$

For joint-pretreated corn stover with DA and ASS, ethanol yield after S-SSF was calculated according to Formula (6) [59]:(6)$$\mathrm{Ethanol}\;\mathrm{yield}\;(\%)=\frac{\mathrm{ethanol}\;\mathrm{in}\;\mathrm{fermentation}\;\mathrm{liquid}\;(g)}{\mathrm{glucan}\;\mathrm{in}\;\mathrm{pretreated}\;\mathrm{corn}\;\mathrm{stover}\left(g\right)\times 1.11\times 0.51}\times 100$$

## 5. Conclusions

Two different pretreatment strategies were proposed to enhance cellulosic ethanol production from corn stover and fractionate chemical compositions of corn stover for biorefinery. It was found that the DA-ASS process could effectively separate and recover hemicellulose and lignin in corn stover for their further application, and the xylose recovery and lignin removal reached 93% and 85%, respectively. The AS process could effectively separate lignin while obtaining high recovery of glucan and xylan. After pretreatment using the two processes, the enzymatic digestibility of pretreated corn stover was effectively improved. Using a fed-batch S-SSF process, the final concentration of ethanol in fermentation liquid reached 48.5 g/L and 50 g/L at a substrate content of 18% with the DA-ASS and AS-pretreated corn stover, respectively. The study provided a reference for cellulosic ethanol production and the biorefinery of lignocellulose. In order to further improve the economic feasibility of corn stover biorefinery, further work will focus on the optimizations of the cellulase system and fermentation process and lignin component utilization.

## Figures and Tables

**Figure 1 ijms-23-13163-f001:**
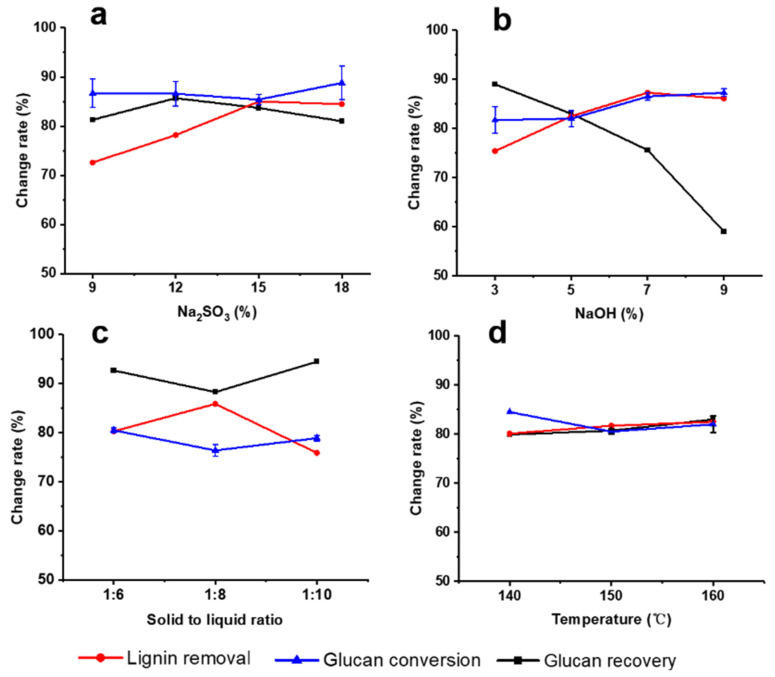
Effects of different ASS pretreatment conditions on lignin removal (circle), glucan recovery (square), and glucan conversion at 72 h of enzymatic hydrolysis (triangle) of DA-pretreated corn stover. (**a**) Na_2_SO_3_ dosages; (**b**) NaOH dosages; (**c**) solid-to-liquid ratio; (**d**) temperature. Here, Na_2_SO_3_ and NaOH were simultaneously added to the pretreatment liquid.

**Figure 2 ijms-23-13163-f002:**
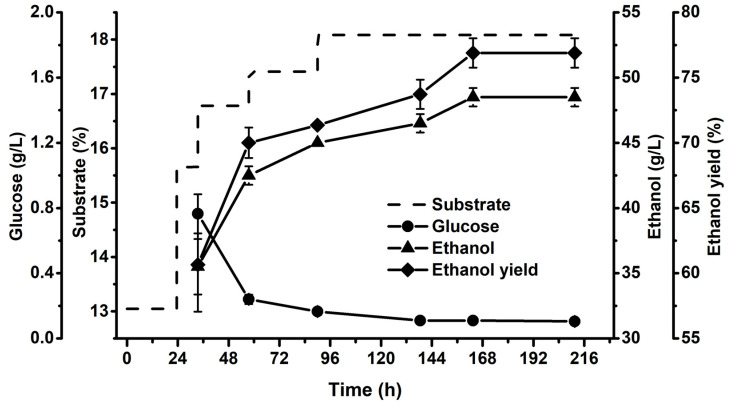
Changes in ethanol yield and concentrations of substrate, glucose, and ethanol during fed-batch S-SSF of DA-ASS-pretreated corn stover using Angel *S. cerevisiae*. Substrate: dotted line; Glucose: circle; Ethanol: triangle; Ethanol yield: rhombus.

**Figure 3 ijms-23-13163-f003:**
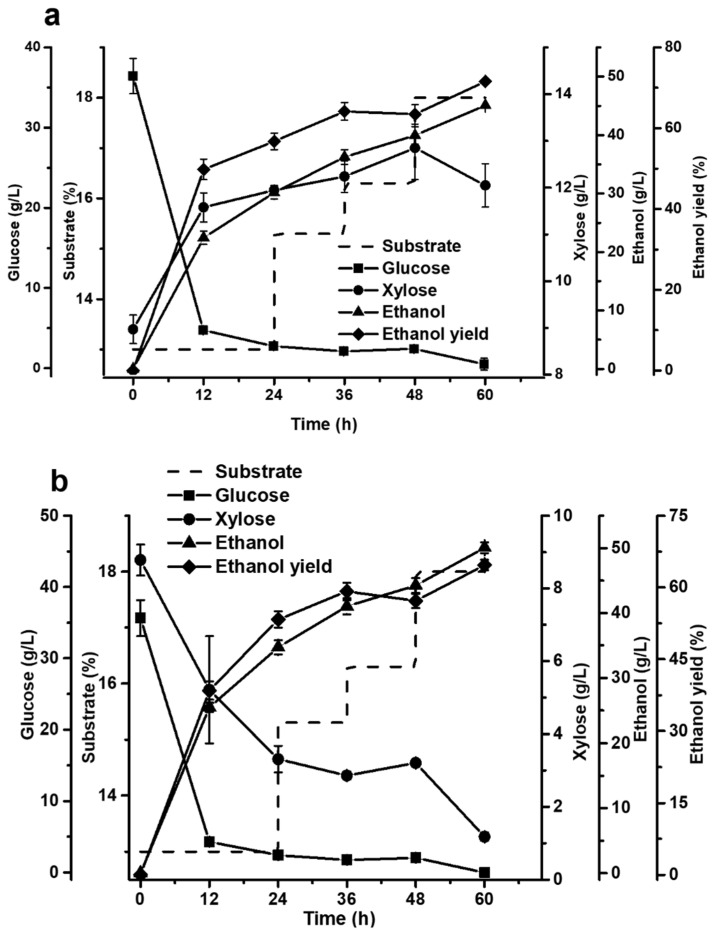
Changes in ethanol yield and concentrations of substrate, glucose, xylose, and ethanol in fed-batch S-SSF of AS-pretreated corn stover using Angel *S. cerevisiae* (**a**) and *S. cerevisiae* XH7 (**b**). Substrate: dotted line; Glucose: square; Xylose: circle; Ethanol: triangle; Ethanol yield: rhombus.

**Figure 4 ijms-23-13163-f004:**
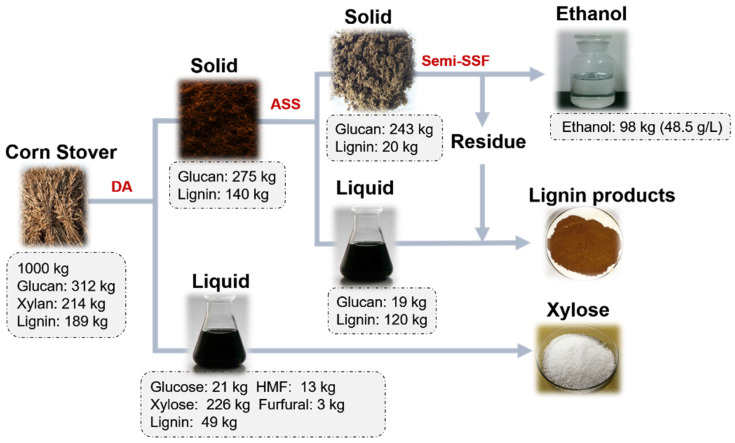
The sketch of the biorefinery of corn stover by joint pretreatment of DA and ASS and preliminary mass balance of the overall process.

**Table 1 ijms-23-13163-t001:** Effects of dilute acid pretreatment on chemical compositions of corn stover, xylan recovery, inhibitors formation, and glucan conversions at 72 enzymatic hydrolysis.

Temperature (°C)	Solid-to-Liquid Ratio	Acid Concentration (%)	Pretreated Solid			Pretreatment Liquid			Glucan Conversion of Enzymatic Hydrolysis (%)
			Glucan (%)	Xylan (%)	Lignin (%)	Xylan Recovery (%)	Furfural (g/L)	HMF (g/L)	
Raw material			31.2 ± 0.6	21.4 ± 0.3	18.9 ± 0.7				24.2 ± 1.0
140	1:8	1.0	46.0 ± 1.0	9.0 ± 0.4	30.1 ± 0.3	53.6	0.010	0.043	64.6 ± 0.8
	1:10	0.8	47.6 ± 0.8	5.5 ± 0.1	30.5 ± 0.5	85.4	0.015	0.051	66.2 ± 3.4
		1.0	46.9 ± 0.4	5.0 ± 0.6	32.0 ± 0.7	85.0	0.016	0.056	68.7 ± 0.7
		1.2	46.6 ± 0.5	3.5 ± 0.3	34.9 ± 1.0	84.8	0.024	0.077	72.4 ± 0.7
150	1:8	1.0	48.2 ± 1.1	5.6 ± 0.5	32.3 ± 0.7	76.8	0.034	0.168	73.9 ± 2.1
	1:10	0.8	47.2 ± 0.3	3.0 ± 0.8	33.5 ± 0.1	93.0	0.026	0.134	71.6 ± 1.9
		1.0	47.7 ± 0.5	2.9 ± 0.4	31.5 ± 1.2	88.1	0.036	0.174	72.6 ± 0.7
		1.2	46.7 ± 0.9	3.1 ± 0.6	34.7 ± 0.2	77.2	0.039	0.203	76.7 ± 2.2

Note: (1) Pretreatment time 50 min; (2) Enzymatic hydrolysis conditions: 20 FPU/g substrate, 50 °C, 150 rpm for 72 h at pH 4.8.

**Table 2 ijms-23-13163-t002:** Changes in chemical compositions and conversions of glucan and xylan in pretreated corn stover before and after AS pretreatment under different conditions.

(NH_4_)_2_SO_3_ (%)	Solid-to-Liquid Ratio	Glucan (%)	Xylan (%)	Lignin (%)	Lignin Removal (%)	Glucan Conversion * (%)	Xylan Conversion * (%)
Raw material		36.6 ± 0.3	21.2 ± 0.6	14.6 ± 0.8		34.6 ± 2.0	38.8 ± 2.6
8	1:6	49.5 ± 0.8	12.3 ± 0.7	10.6 ± 0.6	49.9	73.9 ± 1.3	93.5 ± 5.1
10	1:6	49.0 ± 0.5	12.4 ± 0.9	8.7 ± 0.3	60.9	88.0 ± 2.6	99.8 ± 0.9
12	1:6	50.4 ± 0.4	13.2 ± 0.2	6.7 ± 0.9	69.0	86.6 ± 1.3	99.2 ± 2.3
14	1:6	53.5 ± 1.2	13.5 ± 0.5	3.7 ± 0.5	83.8	87.4 ± 0.1	99.9 ± 4.1
14	1:2	46.7 ± 0.6	11.2 ± 0.4	8.4 ± 1.0	55.8	70.3 ± 1.5	95.9 ± 2.3
14	1:4	52.1 ± 0.8	12.0 ± 0.9	4.7 ± 0.7	80.0	78.5 ± 0.0	95.0 ± 3.0
14	1:8	54.7 ± 0.9	12.4 ± 1.0	4.0 ± 0.6	83.6	75.7 ± 1.2	95.9 ± 1.8

Note: (1) Other pretreatment conditions: 3% (NH_4_)_2_CO_3_, reaction time 1 h. (2) “*”: Enzymatic hydrolysis time 72 h.

## Data Availability

Data included in this study could be available upon request by contacting the corresponding author Jian Zhao via email: zhaojian@sdu.edu.cn.

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
