# Peer review of "Pretreatment Strategies to Enhance Enzymatic Hydrolysis and Cellulosic Ethanol Production for Biorefinery of Corn Stover"

_ijms, 2022, doi:10.3390/ijms232113163_

Round 1
Reviewer 1 Report
Title: Using different pretreatment strategies to enhance enzymatic hydrolysis of corn stover for cellulosic ethanol production and biorefinery
Authors: Wang Sun, Xuezhi Li, Jian Zhao, Yuqi Qin
Journal: International Journal of Molecular Sciences
Title: Though the title adequately conveyed the content of the article, it can be constructed better.
I suggest: Pretreatment strategies to enhance ethanol production from Corn Stover through enzymatic hydrolysis
Abstract: The first sentence of the abstract should be rewritten. The motivation, aim, study objective should be mentioned. The (AS) in Ln 20 should be deleted. The application of the result should be included.
Introduction
Ln 28-32. The four lines sentence should be rewritten or divided into two.
Ln 69 should read: For example, Li et al [27]..........
Ln 74 should read: Qi et al. [28] used..........
Ln 86: replace manuscript with study
The introduction section is incomplete and deficient. The authors should add a paragraph to review previous published works in this area and the outcome.
The authors can include data on bioethanol demand/production/consumption in their country or worldwide. This will show why further research in bioethanol production is needed. Also, the research question, motivation for study, study aim, objectives, scope, and expected applications of the outcome of the study are missing or not clearly stated.
Results and discussion
Well presented and clearly written
Materials and Methods
Ln 337-338: ‘certain proportion and reacted for a required time at the set temperature up’ should be clarified
Conclusions
Ln 402-406. Too long sentence.
The Conclusion is inadequate in its present form. Authors should improve it. Authors should include any challenges encountered during the experiment. Authors should include the applications of the study outcomes and recommend areas of further study.
Generally, the articles are well written with significant technical content. The Tables and Figures were well presented and unambiguous. The literature cited were adequate, relevant and recent.
However, there are few omissions and spelling mistakes that should be corrected The study will be of interest to scholars in biomass conversion and renewable energy. It will advance the production of bioethanol from wastes and contribute to biorefinery.
Reviewer 2 Report
Comments to the Authors
This work is therefore a useful contribution to the area of bioethanol production of from plant biomass.
The work is generally legible and written with great care.
However, please see the following notes:
- line 104 - please change "the liquid ratio" to "the solid-liquid ratio"
- 2.1.2. Effect of ASS pretreatment - Please explain carefully whether treatment with NaOH was performed first and then with Na2CO3? Or at the same time?
- line 162 - Table 2 - please change "Xyaln" to "Xylan"
- 2.3.2. Fed-batch S-SSF of AS pretreated corn stover - make it clear that Angel S. cerevisiae was used for comparison only
- 4.3. Enzymatic hydrolysis - Please state if this concerns prehydrolysis prior to SSF process?
Reviewer 3 Report
The article has new results for the production of bioethanol from bio-based feedstocks. It has also new results on the fractionation of feedstocks in different compounds and treatment of by-products which have value, but were not so well used in previous studies. Before this article will be accepted, it is necessary that the improvements will be made:
(1) Very limited information in the introduction on the background of this research. There are so many articles on changes in operational conditions that it is a pity that author does not clearly state what he has modified to use this specific method as a main method for the fractionation. More articles from EU and Nord Amerika should be integrated.
(2) Methodology does not contain any info on statistics and design of experiments.
(3) Results have too many tables and images so that it would be more beneficial to combine some images and make it more applicable for the visualization.
(4) Discussion does not show the uniqueness of this article and its socioeconomic impact.
(5) Conclusion is short, but it does not show what novel the authors have done in this work.
Reviewer 4 Report
Please check the title row of table 2 for typo.
Please explain why further increase of ammonium sulfite dosage did not result in a higher conversion.
Please provide a table presenting the effect of the pretreated corn stovers on the cell growth
Do you have any reference for formula you have used to calculate the recovery, yield, and removal %?
Round 2
Reviewer 3 Report
all comments were integrated